# “Double Cross Sign” Could Be an Indicator of an Adequate Amount of Bone Cement in Kyphoplasty with the SpineJack System: A Retrospective Study

**DOI:** 10.3390/diagnostics12123068

**Published:** 2022-12-06

**Authors:** Chao-Jui Chang, Chih-Kai Hong, Che-Chia Hsu

**Affiliations:** 1Departments of Orthopedics, National Cheng Kung University Hospital, College of Medicine, National Cheng Kung University, Tainan 70403, Taiwan; 2Department of Surgery, National Cheng Kung University Hospital, College of Medicine, National Cheng Kung University, Dou-Liou Branch, Douliu 64043, Taiwan

**Keywords:** compression fracture, kyphoplasty, SpineJack, bone cement amount, functional outcome

## Abstract

Kyphoplasty with the SpineJack system was able to restore vertebral height and stabilize the vertebra with an injection of bone cement. The goal of this study was to seek a reliable assessing method during the surgery in determining the minimum amount of bone cement required for the SpineJack system to restore vertebral height and stabilize the vertebra. We defined the “double cross sign” as bone cement that expanded vertically along the bilateral SpineJack system, and spread across the midline of the vertebral body as viewed in the anteroposterior (AP) view of the radiographic image. Sixty-five patients aged 74.5 ± 8.5 years with vertebral compression fracture were included in the study. Patients with a positive double cross sign had better ODI score than those without the double cross sign (20.0 ± 6.9 vs. 32.3 ± 8.2; *p* < 0.001). Postoperative regional kyphotic and local kyphotic angle were significantly better in the positive double cross sign group (11 ± 8.8 degrees vs. 5.3 ± 3.2 degrees; *p* = 0.001/11.7 ± 6.2 degrees vs. 6.6 ± 4.1 degrees; *p* = 0.001, respectively). The more stable construct was built once the double cross sign was achieved during surgery. In this study, a convenient and intuitive method in identifying the minimum but sufficient quantity of injected cement during the SpineJack procedure was developed.

## 1. Introduction

Osteoporosis and its associated age-related fractures constitute a major public health problem [1,2]. Due to the increasing lifespan of the world’s population, osteoporotic vertebral compression fractures are becoming increasingly prevalent [3]. The risk of an additional fracture increases up to fivefold once a vertebral fracture has occurred [4]. For example, in the United States, the cost of treatment for incident osteoporotic fractures was an estimated USD 17 billion in 2005 [5]. A study conducted in China, home to 20% of the world’s elderly population, also reported a cost for treatment of vertebral compression fractures of more than USD 3000 per patient [6].

In patients with vertebral compression fractures, nonoperative treatment is often difficult to tolerate. Both vertebroplasty and kyphoplasty are well-accepted minimally invasive procedures for the management of painful osteoporotic compression fractures [7,8,9]. Although adverse events are rare with both techniques, higher rates of procedure-related complications such as cement leakage were noted in vertebroplasty [10], and a higher incidence of an adjacent fracture was found after balloon kyphoplasty [11]. The SpineJack is a newly introduced minimally invasive technique. SpineJack is a titanium implant designed to restore the original vertebral height, and to stabilize the collapsed vertebra with an injection of bone cement [12]. It is inserted into the vertebral body with a bilateral transpedicular minimally invasive approach. After insertion, it gradually expands similar to a small car jack and provides a constant distraction force. Even spreading of cement (polymethyl methacrylate (PMMA)) can be achieved with two symmetrically positioned devices inside the vertebral body, and thus may reduce the risk of cement leakage.

To stabilize the desired height reduction, the entire cavity should be filled with cement. However, controversy exists regarding the minimum amount of cement that should be used to achieve long-term restoration, which is essential to minimize complications [13,14,15,16,17]. Therefore, this retrospective observational study sought to identify the minimum quantity of cement required while sufficiently stabilizing the restored fractured vertebra using the SpineJack system. We hypothesized that once the “double cross sign” was observed during the surgery, a more stable construct could be created, which may lead to better functional outcome.

## 2. Materials and Methods

### 2.1. Population

This retrospective study examined 100 consecutive patients who suffered from thoracic or lumbar vertebral compression fracture (VCF) and underwent an operation using SpineJack by the same senior orthopedic spine surgeon from November 2016 to September 2018. VCF was diagnosed and confirmed by either radiograph or computed tomography scan at the out-patient department. Of them, 65 patients met the following criteria and were included in the study: men or women aged between 23 and 88 years with a vertebral compression fracture (VCF) involving the lower thoracic or lumbar vertebrae (between T9 and L4) which occurred <3 months from the operation with a loss of height in the anterior, middle, or posterior third of the vertebral body that was considered unstable (grade A3 according to Magerl’s classification), patients who failed conservative medical therapy, and patients with a target vertebral body suitable for the SpineJack procedure (with a minimum internal pedicle diameter > 5.8 mm to allow placement of the device) [12]. A total of 35 patients who met the following criteria were excluded: target VCFs due to high-energy trauma or suspected tumor, evidence of unstable neurologic deficit on physical examination, radiographic evidence of pedicle fracture, lost to follow-up, and history of spine surgery that included prior vertebral augmentation or interbody fusion. The Appendix A data included in this retrospective observational study were collected at two clinical sites in Taiwan by a single senior orthopedic surgeon (National Cheng Kung University Hospital (NCKUH)/Ministry of Health and Welfare Tainan Hospital).

National Cheng Kung University Hospital (NCKUH) has approved the waiver for informed consent in the current study and the institutional review board (IRB) number was B-ER-109-450. The IRB of NCKUH is operated according to the laws of ICH-GCP and of Central Competent Authorities. All experimental protocols were approved by an IRB of NCKUH. All methods were carried out in accordance with relevant guidelines and regulations.

### 2.2. Preoperative Assessment

All patients underwent a complete preoperative clinical examination that included a detailed medical history and complete radiographic examination to confirm the presence, location, and severity of the VCF. Occasionally, a computed tomography scan was used to assess structural deformities before the procedure. Anterior, middle, and posterior vertebral body heights were recorded directly in the sagittal sections. The regional kyphotic angle was defined as the angle between the upper endplate of T11 and the lower endplate of L3. The local kyphotic angle was defined as the angle between the upper and lower plates of the fractured vertebra.

### 2.3. Operation

All surgeries were performed by single senior orthopedic spine surgeon (C-C H) according to standard procedures. Most procedures were conducted under local anesthesia. The patient was placed in a prone position. The SpineJack was inserted in an unexpanded condition with the percutaneous or minimally invasive posterior surgical approach using the surgical tools supplied with the system.

After insertion into the vertebral body, the device was expanded with a designated tool from the SpineJack instrumentation kit, which locked the device and pulled the two axial ends of the implant toward each other. Longitudinal compression of the device caused the implant to open only in the cranio-caudal direction due to the machined grooves. Once SpineJack reached the desired expansion and the reduction was achieved, the device was left in place inside the restored vertebra and high-viscosity PMMA bone cement was injected into and around the implant. Real-time fluoroscopic monitoring with biplane images obtained by rotating the C-arm throughout the device insertion, expansion, and cement injection ensured the correct implantation [18,19,20].

### 2.4. Postoperative Assessment

Patient-reported outcomes were evaluated 5.6 ± 3.5 months postoperatively during the out-patient department visit. The main outcome was daily functional capacity assessed with the self-administered Oswestry Disability Index (ODI). Complications such as subsequent adjacent or nonadjacent fractures, cement leakage into the extravertebral space, device-related adverse incidents, and surgery-related complications were recorded throughout the follow-up period.

Image assessment that included anterior, middle, posterior vertebral body heights, and regional and local kyphotic angles was carried out before and after the operation (Figure 1). We defined the “double cross sign” as bone cement that expanded vertically along the bilateral SpineJack system and across the midline of the vertebral body in the anteroposterior (AP) view of the radiographic image (Figure 2). The assessment of the postoperative radiographic image that met the criteria for double cross sign was also conducted and recorded by the same single senior orthopedic spine surgeon (C-CH). In addition, the angle between the two expanded devices was recorded immediately after the surgery, and at the latest postoperative radiographic follow-up (Figure 3).

### 2.5. Statistical Analysis

Statistical comparisons were conducted with IBM SPSS Statistics for Apple Macintosh OSX (Version 25; IBM SPSS Inc.; New York, NY, USA) The Mann–Whitney U test was performed to identify differences in age, ODI, bone mineral density (BMD), injected cement amount, and changes in the kyphotic angle and vertebral body height between the groups. Statistical significance was accepted for *p* values < 0.05.

## 3. Results

Patient characteristics are summarized in Table 1. A total of 65 patients met the inclusion criteria. Patients (76.9% women) were aged 74.5 ± 8.5 years (range, 53 to 88 years). The mean follow-up period was 5.6 months. Adjacent level fractures were found in 11 patients during their clinical follow-up. Thirteen patients experienced minor PMMA cement leakages without any clinical consequence (seven patients had intradisk leakage and six patients had prevertebral leakage). The leakages were all asymptomatic with no sequelae on the clinical outcome.

The operative characteristics of the patients with and without a postoperative double cross sign are shown in Table 2. The average volume of cement injected around the devices was 7.3 mL. The average injected volume of cement in the patients with a double cross sign was 7.2 ± 2.8 mL (Table 1). No worsening of posterior wall protrusion was noted. No major procedure-related complications occurred in the immediate postoperative period, and no device had to be removed.

We obtained complete functional capacity as assessed by the ODI during the out-patient department follow-up. Patients with a positive double cross sign had lower ODI score than those without the double cross sign (20.0 ± 6.9 vs. 32.3 ± 8.2; *p* < 0.001). Significantly better postoperative regional kyphotic angle correction was noted in the positive double cross sign group (11 ± 8.8 degrees vs. 5.3 ± 3.2 degrees; *p* = 0.001). Postoperative local kyphotic angle correction was significantly increased in patients with the double cross sign (11.7 ± 6.2 degrees vs. 6.6 ± 4.1 degrees; *p* = 0.001). Anterior vertebral body height change was significantly higher when the double cross sign was achieved during the surgery (155 ± 159% vs. 40 ± 47%; *p* < 0.001). Patients with the double cross sign also experienced more middle vertebral body height change after the operation compared with those without the double cross sign (156 ± 132% vs. 84 ± 82%; *p* = 0.012). The positive double cross sign group had significantly less angle change between the two inserted devices than in those without the double cross sign (3.7 ± 4.3 degrees vs. 9 ± 4.7 degrees; *p* = 0.002). A more stable construct could be achieved once the double cross sign was created during the operation.

## 4. Discussion

The SpineJack system has been shown to stabilize the collapsed vertebra and correct kyphosis, but consensus has not been reached regarding the amount of cement to inject. The most important finding of this study was that once the double cross sign was observed during the SpineJack procedure, a better functional capacity (ODI) was shown in the patient’s short-term postoperative follow-up. Better corrections in the regional and local kyphotic angles were observed when the double cross sign was seen during the surgery. Additionally, more anterior and middle vertebral body height changes were found in patients with the double cross sign than in patients without the double cross sign. The strength of this study is that we found a convenient and intuitive intraoperative evaluation to identify the minimum quantity of cement necessary while sufficiently stabilizing the restored fractured vertebra with the SpineJack procedure.

Height restoration and vertebral kyphosis angle reduction could be successfully achieved by the kyphoplasty, which could also be beneficial for urinary infection prevention, weight loss avoidance and reduction of pulmonary diseases and mortality risks [21,22,23]. A previous study showed that the SpineJack system demonstrated a higher potential for vertebral body height restoration and maintenance over time compared with balloon kyphoplasty [24]. In the current study, kyphotic angle and vertebral body height were reduced after the SpineJack procedure. In addition to the deformity correction, the present study also found that there was improvement in daily functional capacity, which was consistent with previous clinical studies [19,25,26].

Risks of cement leakage and associated complications have been crucial issues in vertebroplasty and kyphoplasty. A reduced amount of cement may be ineffective, while larger volumes may leak outside the border of the vertebral body and thus increase the incidence of neurologic complications [27]. Taylor et al. found that some cement leakage led to pulmonary embolism, nerve root pain or radiculopathy [10]. Injected cement may leak into a variety of anatomical compartments as follows: prevertebral soft tissue (6~52.5%), spinal canal (37.5%), intervertebral disk (5~25%), prevertebral veins (5~16.6%), epidural veins (16.5%), as well as the inferior vena cava and lungs [28,29,30,31]. Similar to the incidence rates of previous studies, 13 patients (20%) in our study experienced asymptomatic cement leakage events.

New adjacent vertebral body fractures associated with cement augmentation have been reported. A cadaveric study showed that for a reduction in the risk of upper adjacent vertebral fracture, it was better to restore the height of the injured vertebral body, and to decrease the angle of kyphosis [32]. In one finite element model performed by Ottardi et al., they report that kyphoplasty rather than vertebroplasty is preferred due to the ability to restore the initial vertebral body height, reduce the stresses on the adjacent endplates and decrease the risk of fracture [33]. Kévin et al. found secondary adjacent level fractures in 21.1% of patients who underwent the SpineJack procedure [25]. In our study, adjacent level fractures were found in 11 patients (17%) during their postoperative follow-up. Overall, the achievement of the double cross sign with the SpineJack system did not give rise to the occurrence of cement leakages (16%) and adjacent level fractures (16%).

The amount of bone cement that should be injected into the vertebral body during vertebroplasty and kyphoplasty has remained controversial. Martin et al. reported that an injection of 4~6 mL of PMMA was effective [14]. However, this volume may not be universally suitable due to the variability of the vertebral body volume among the different sexes or races. Jin et al. conducted a volumetric analysis of cement in vertebroplasty and showed that a volume larger than 11.65% led to a significantly increased incidence of cement leakage and adjacent fractures [15]. A biomechanical study that compared the SpineJack device and balloon kyphoplasty to treat traumatic fractures revealed that kyphoplasty required a 30% cement volume, while it was possible to reduce the amount of cement to 10% of the vertebral body volume with the use of SpineJack without compromising the reposition height after reduction [16]. Another clinical analysis of cement volume and distribution in kyphoplasty found that the diagnostic value of cement distribution was better than that of cement volume for relieving patient pain. They also recommended that the cement distribution value should be above 0.49 [17]. In our present study, the average injected volume of cement in the patients with double cross sign was 7.7 ± 2.9 mL. Although the amount of bone cement injected in the group with double cross sign was not significantly different compared with the group without the double cross sign, patients had a better outcome in both aspects of daily function and radiographic image findings when double cross sign had been seen in the operation.

This study had several limitations. This was a retrospective observational study design, with a short follow-up period. Additionally, the improvement of daily functional capacity after the SpineJack procedure between the groups with and without the double cross sign was not clear because a preoperative ODI score was not obtained for our enrolled patients. The current study had found that the achievement of double cross sign did not demand significantly more bone cement than in the negative group. In spite of a similar injected bone cement amount, better corrections in kyphotic angles and more vertebral body height changes were noticed in the double cross sign group. Further studies can focus on whether intraoperative double cross sign will lead to better outcome in different surgeries such as vertebroplasty or balloon kyphoplasty.

## 5. Conclusions

Our findings suggest that the ideal minimum quantity of injected cement in the SpineJack system may depend on the presence of an intraoperative double cross sign. The more stable construct was built once the double cross sign was created during the surgery.

## Figures and Tables

**Figure 1 diagnostics-12-03068-f001:**
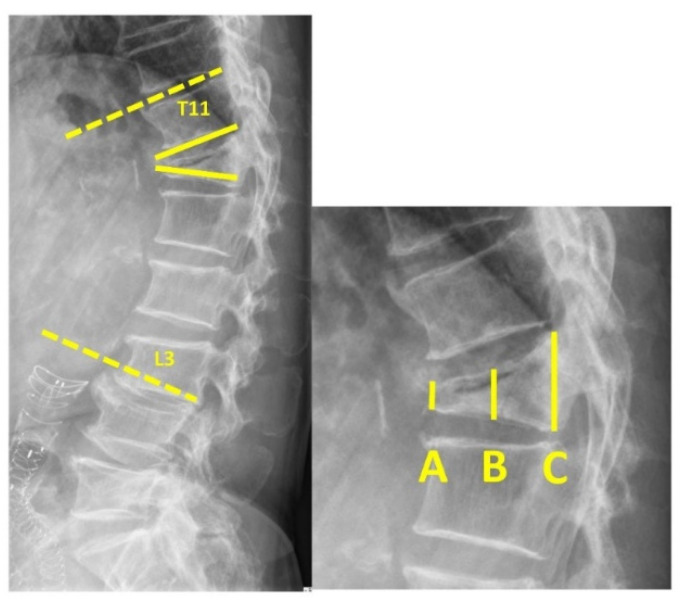
(**Left**) Regional kyphotic angle—the angle between the upper endplate of T11 and lower endplate of L3 (dashed line). Local kyphotic—the angle between the upper and lower endplates of the fractured vertebra (solid line). (**Right**) A—anterior VBH, B—middle VBH, C—posterior VBH (VBH = vertebral body height).

**Figure 2 diagnostics-12-03068-f002:**
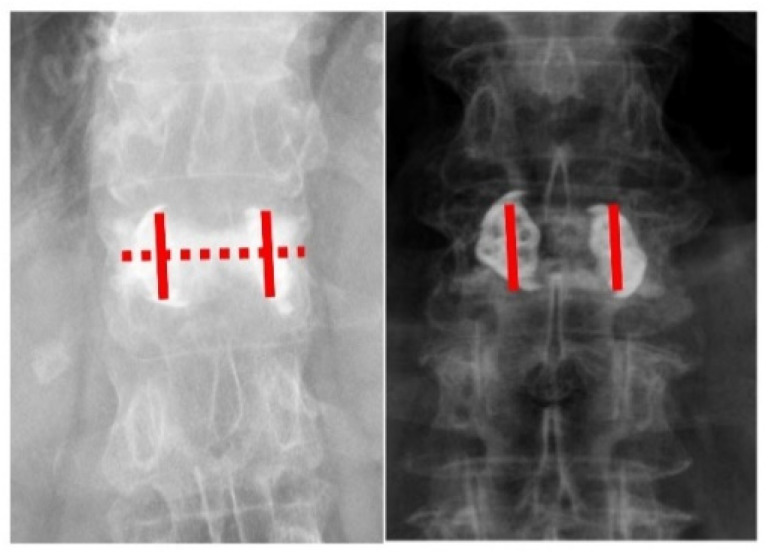
(**Left**) The AP view of the radiographic image shows a positive double cross sign. The bone cement expands vertically along the bilateral SpineJack system, and across the midline of the vertebral body. (**Right**) The AP view of the radiographic image shows a negative double cross sign. The bone cement expands vertically along the bilateral SpineJack system without spreading across the midline of the vertebral body.

**Figure 3 diagnostics-12-03068-f003:**
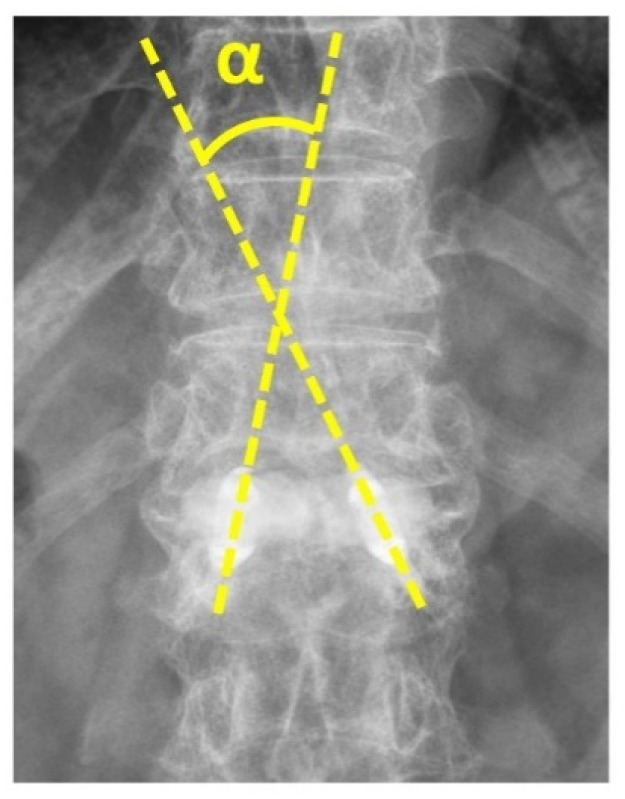
The AP view of radiographic image shows the angle between the two expanded devices (α angle).

**Table 1 diagnostics-12-03068-t001:** Baseline patient characteristics.

Characteristics	
Patients (number)	n = 65
Age (years)	74.5 ± 8.5
Women n (%)	50 (76.9)
Follow-up time (months)	5.6 ± 3.5
Operated level n (%)	
T9	1 (1.5%)
T10	1 (1.5%)
T11	2 (3%)
T12	17 (26.2%)
L1	21 (32.3%)
L2	11 (16.9%)
L3	6 (9.2%)
L4	6 (9.2%)
Magerl classification n (%)	
A3.1	29 (45%)
A3.2	26 (40%)
A3.3	10 (15%)
Cement leakage	n = 13 (20%)
Adjacent level fracture	n = 11 (16.9%)
Injected cement (mL)	7.3 ± 2.7

**Table 2 diagnostics-12-03068-t002:** Operative characteristics of patients with and without the double cross sign.

	Positive Double Cross Sign (n = 45)	Negative Double Cross Sign (n = 20)	*p* Value
Age (years)	75.5 ± 7.3	72.4 ± 10.6	*p* = 0.250
Follow-up (months)	6.2 ± 3.6	4.4 ± 3.1	*p* = 0.115
Oswestry Disability Index	20.0 ± 6.9	32.3 ± 8.2	*p* < 0.001 *
Lumbar spine BMD (g/cm^2^)	0.86 ± 0.1	0.91 ± 0.1	*p* = 0.333
Lumbar spine T-score	−2.3 ± 1.2	−1.7 ± 1.0	*p* = 0.153
Injected cement amount (mL)	7.7 ± 2.9	6.6 ± 2.3	*p* = 1.035
Δ regional kyphotic angle (degrees)	11 ± 8.8	5.3 ± 3.2	*p* = 0.001 *
Δ local kyphotic angle (degrees)	11.7 ± 6.2	6.6 ± 4.1	*p* = 0.001 *
Δ anterior VBH (%)	155 ± 159	40 ± 47	*p* < 0.001 *
Δ middle VBH (%)	156 ± 132	84 ± 82	*p* = 0.012 *
Δ posterior VBH (%)	15 ± 19	14 ± 12	*p* = 0.832
Angle change between devices (mL)	3.7 ± 4.3	9 ± 4.7	*p* = 0.002 *
Cement leakage (mL)	7	6	0.428
Adjacent fracture n	7	4	0.805

BMD = bone mineral density; VBH = vertebral body height; regional = T11-L3 level; Δ = difference in the measurement before and after the surgery. All data are reported as mean ± standard deviation. * *p* value < 0.05.

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
