# Peer review of "“Double Cross Sign” Could Be an Indicator of an Adequate Amount of Bone Cement in Kyphoplasty with the SpineJack System: A Retrospective Study"

_diagnostics, 2022, doi:10.3390/diagnostics12123068_

Round 1
Reviewer 1 Report
The authors present here a retrospective study on the use of an intravertebral distractible implant in combination with intravertebral cement augmentation. The structure of the study and the presentations are basically good. Unfortunately, there is a glaring error of interpretation in the results:
The ODI score has the following internationally recognised structure: the best value a patient can achieve without any complaints is zero, and the worst value is 100! I personally have not come across an inverse representation of this score in more than 1000 international reviews.
This means that the values of 80.3 described here in the group with the discussed "double cross sign" would be almost bedridden and the study results would be considered catastrophic.
I see a massive problem here! A publication is submitted in which the authors do not realise that the ODI value was misunderstood or incorrectly documented and understood. This raises the legitimate question of why the authors do not notice such a thing and whether this data was actually ever collected! In this day and age of increasing falsified study data, all alarm bells are ringing at this point.
Irrespective of this publication, this requires a comprehensible explanation from all authors involved, so that they do not disqualify themselves scientifically...
All other comments are, in my view, unimportant at this point.
Reviewer 2 Report
Reviewer Comments
Thank you very much for the opportunity to review the manuscript submission entitled: “Double cross sign” could be an indicator of an adequate amount of bone cement in kyphoplasty with the Spine Jack system: a retrospective study.
The current study aims to determine the minimum amount of bone cement required for the Spine Jack system to restore vertebral height and stabilize the vertebra.; The study is interesting; however, some limitations and constructive comments are pointed out below:
Specific comments
The manuscript must be proofread for grammatical errors by a native English speaker.
Title and abstract:
· The objective and conclusion should be in line.
· Mention the diagnosis of the patients.
· End the abstract with the clinical significance of the study
· Include MeSH terms as keywords.
Introduction
· There are a lot of sentences that require referencing.
o Osteoporosis and its associated age-related fractures constitute a major public health problem.
o For example, in the United States, the cost of treatment for incident osteoporotic fractures was an estimated $17 billion in 2005
o In patients with vertebral compression fractures, nonoperative treatment is often difficult to tolerate. Both vertebroplasty and kyphoplasty are well-accepted minimally invasive procedures for the management of painful osteoporotic compression fractures.
o Like this, look for all the sentences.
· Explain the scientific background and rationale for the investigation being reported.
· State-specific prespecified hypotheses
Methods
· Present key elements of study design early in the paper
· Clearly describe the diagnostic criteria.
· Describe the setting, locations, and relevant dates, including periods of recruitment and data collection.
· Give the clear eligibility criteria, and the sources and methods of selection of participants.
· Most of the methods section the references are missing. The inclusion criteria, the surgical methods utilized needs referencing.
· Clearly Explain how the study size was arrived at.
· Justify the use of statical tests used. Did the data follow normal distribution?
Discussion
· Give a cautious overall interpretation of results considering objectives, limitations, multiplicity of analyses, results from similar studies, and other relevant evidence
· Discuss the generalisability (external validity) of the study results
Round 2
Reviewer 1 Report
Unfortunately, the revised version of this publication must also be rejected from a scientific point of view. The authors have not understood the implications of their "misunderstanding" of the ODI value documentation.
Simply reversing the figures in a revised version is absolutely not a scientific approach. I am frankly appalled by this approach, as I thought I had made it sufficiently clear in the first round of review that the data presented here had lost its credibility. Turning over the same data and submitting the same paper again makes me doubt the authors' basic understanding of scientific work.
I will not be able to accept this work as scientifically acceptable.
Reviewer 2 Report
The authors have satisfactorily addressed all my comments. It can be accepted for publication.